# Structural basis for antiepileptic drugs and botulinum neurotoxin recognition of SV2A

Atsushi Yamagata ⊙[1] ✉, Kaori Ito[1], Takehiro Suzuki ⊙[2], Naoshi Dohmae ⊙[2], Tohru Terada ⊙[3] & Mikako Shirouzu ⊙[1]

More than one percent of people have epilepsy worldwide. Levetiracetam (LEV) is a successful new-generation antiepileptic drug (AED), and its derivative, brivaracetam (BRV), shows improved efficacy. Synaptic vesicle glycoprotein 2a (SV2A), a putative membrane transporter in the synaptic vesicles (SVs), has been identified as a target of LEV and BRV. SV2A also serves as a receptor for botulinum neurotoxin (BoNT), which is the most toxic protein and has paradoxically emerged as a potent reagent for therapeutic and cosmetic applications. Nevertheless, no structural analysis on AEDs and BoNT recognition by full-length SV2A has been available. Here we describe the cryo-electron microscopy structures of the full-length SV2A in complex with the BoNT receptor-binding domain, BoNT/A2 H$_C$, and either LEV or BRV. The large fourth luminal domain of SV2A binds to BoNT/A2 H$_C$ through protein-protein and protein-glycan interactions. LEV and BRV occupy the putative substrate-binding site in an outward-open conformation. A propyl group in BRV creates additional contacts with SV2A, explaining its higher binding affinity than that of LEV, which was further supported by label-free spectral shift assay. Numerous LEV derivatives have been developed as AEDs and positron emission tomography (PET) tracers for neuroimaging. Our work provides a structural framework for AEDs and BoNT recognition of SV2A and a blueprint for the rational design of additional AEDs and PET tracers.

Levetiracetam (LEV), approved by the Food and Drug Administration in 1999, is a new generation of AEDs[1]. LEV is the most successful AED because of its broad spectrum of activity, long retention, rapid absorption, and good tolerability. Synaptic vesicle glycoprotein 2A (SV2A) has been identified as the brain target of LEV[2,3]. LEV strictly targets SV2A within three members of the SV2 family: SV2A, SV2B, and SV2C[2]. Brivaracetam (BRV), a LEV derivative, is an approved AED with a 15- to 30-fold higher affinity for SV2A, high brain permeability, and fast onset of action[4–6]. SV2A is exclusively expressed in neurons and is present in all brain regions[7,8]. The wide distribution of SV2A in almost all neuron types has led to the development of various LEV-based

radioactive positron emission tomography (PET) tracers for neuroimaging[9,10]. In particular, [$^{11}$C]UCB-J has been widely used and dramatically improved synaptic density imaging[11] of neurodegenerative diseases, including Alzheimer's disease and Parkinson's disease, epilepsy, schizophrenia, aging, and viral infection[10,12–14].

To date, one homozygous mutation and two heterozygous mutations in SV2A have been identified in patients with epilepsy, who showed poor response to LEV and/or additional or increased seizures after treatment with LEV[15–17]. SV2A knock-out (KO) mice experience severe seizures and die within three weeks[18]. Neurons from SV2A/SV2B double knock-out (DKO) mice show an abnormal increase in

[1]Laboratory for Protein Functional and Structural Biology, RIKEN Center for Biosystems Dynamics Research, 1-7-22 Suehiro-cho, Tsurumi-ku, Yokohama, Kanagawa, Japan. [2]Biomolecular Characterization Unit, RIKEN Center for Sustainable Resource Science, 2-1 Hirosawa, Wako, Saitama, Japan. [3]Department of Biotechnology, Graduate School of Agricultural and Life Sciences, The University of Tokyo, 1-1-1 Yayoi, Bunkyo-ku, Tokyo, Japan. ✉e-mail: atsushi.yamagata@riken.jp

neurotransmitter release, suggesting a fundamental role of SV2A in synaptic transmission[19]. SV2 family proteins belong to the major facilitator superfamily (MFS) with 12 predicted transmembrane helices[20,21]. In addition to the transmembrane domain (TMD) core, a cytosolic N-terminal region and a large fourth luminal domain (LD4) are present. SV2A interacts with synaptotagmin 1 (SYT1) via its N-terminal region in a $Ca^{2+}$-dependent manner[22–24]. SYT1 is a $Ca^{2+}$ sensor during SV exocytosis[25–27], and the interaction between SV2A and SYT1 likely plays an important role in modulating the $Ca^{2+}$-dependent SV release[28]. However, this may explain only part of the SV2A function, because the SV2A mutant lacking its N-terminal cytoplasmic region can restore normal synaptic transmission in neurons cultured from SV2A/B DKO mice[29].

Botulinum neurotoxins (BoNTs), including seven serotypes, BoNT/A-G, are the most potent neurotoxins produced by *Clostridia*[30,31]. Each serotype is further divided into subtypes (e.g., BoNT/A1-A10, BoNT/B1-B8, BoNT/E1-E12, BoNT/F1-F9)[32,33]. They share a common domain architecture consisting of a 100-kDa heavy chain and a 50-kDa catalytic light chain linked by a disulfide bond[34–36]. The catalytic light chain is a $Zn^{2+}$ endopeptidase that cleaves the *N*-ethylmaleimide-sensitive factor attachment protein receptor (SNARE) family protein in presynaptic nerve terminals, preventing the neurotransmitter release[37,38]. FDA-approved BoNT/A1 and BoNT/B1 are now widely used in clinical and esthetic medicines[39,40]. The BoNT heavy chain is further divided into an N-terminal translocation domain ($H_N$) and a C-terminal receptor-binding domain ($H_C$). A dual-receptor mechanism, in which the $H_C$ synergistically targets both ganglioside and protein receptors on the neuronal plasma membrane, enables the exquisite specificity of BoNT and neurons[41]. SV2 and SYT have been identified as BoNT protein receptors. BoNT/A, BoNT/D, BoNT/E, and the related tetanus neurotoxin target the SV2 proteins[42–46], whereas BoNT/B, BoNT/G, and the mosaic toxin BoNT/DC target SYT1 and SYT2[47–50]. To date, the crystal structures of the isolated LD4 domain of SV2C (LD4C) and the SV2A LD4:SV2C LD4 fusion protein (LD4AC) have been determined, showing unique beta-helix structures[51,52]. The structures of the complexes between BoNT/A1 $H_C$ ($H_C$A1) and LD4C in the glycosylated and the unglycosylated states revealed that $H_C$A1 recognizes LD4C through backbone–backbone and protein-glycan interactions[51,53]. The recent structural analysis of the complex between BoNT/E $H_C$ ($H_C$E) and LD4AC exhibited an unexpected binding mode distinct from the $H_C$A–SV2C complex[52]. Importantly, however, no structural information has been available on the interaction between the intact SV2A and BoNT $H_C$.

To elucidate the antiepileptic mechanism of the racetam-based AEDs, such as LEV and BRV, and to provide clues to the molecular mechanism of SV2, we determined the cryo-electron microscopy (cryo-EM) structures of human SV2A in complex with LEV (SV2A–LEV), that in complex with BoNT/A2 $H_C$ (SV2A–$H_C$A2), that in complex with both LEV and BoNT/A2 $H_C$ (SV2A–$H_C$A2–LEV), and that in complex with both BRV and BoNT/A2 $H_C$ (SV2A–$H_C$A2–BRV). Our study provides a structural framework for understanding the recognition and uptake mechanism of racetam derivatives by SV2A and the targeting of SV2A by BoNT.

## Results

### SV2A overall structure

The full-length SV2A (UniprotID: Q7L0J3 residues 2–742) fused with N-terminal FLAG tag was overexpressed in insect (*Spodoptera frugiperda* Sf9) cells and purified in the detergent lauryl maltose neopentyl glycol (LMNG) with cholesterol hemisuccinate (CHS) (Supplementary Fig. 1). The BoNT/A2 $H_C$ ($H_C$A2) was expressed in *Escherichia coli*. Purified $H_C$A2 was mixed with purified SV2A to form a complex, and then LEV was added. We used cryo-EM single-particle reconstruction to determine the structure of the SV2A–$H_C$A2–LEV complex. $H_C$A2 acts as a fiducial marker to facilitate image alignment in 2D classification.

We successfully reconstructed a cryo-EM map of the SV2A–$H_C$A2–LEV complex with a nominal resolution of 2.88 Å (Fig. 1a and Supplementary Figs. 2, 3). The quality of the cryo-EM map is sufficient for modeling the SV2A and $H_C$A2, except that the peripheral region of the $H_C$A2 shows poor density (Supplementary Fig. 3b). Additional masked local refinement improved the map quality covering $H_C$A2 and SV2A LD4 (LD4A) (Fig. 1a and Supplementary Fig. 3b). In addition to the monomeric SV2A–$H_C$A2–LEV complex, we also obtained the cryo-EM reconstruction of the dimeric assembly of the SV2A–$H_C$A2–LEV complex to 2.90 Å resolution, in which two SV2A–$H_C$A2–LEV complexes are assembled through the LD4–LD4 interaction, positioning two SV2A TMDs in a near perpendicular configuration (Supplementary Figs. 2, 4). Although the dimeric assembly invokes its functional relevance (see also "Discussion"), we focus on the monomeric complex hereafter.

The residues 2–136 of SV2A, predicted to be disordered regions, are not visible in the cryo-EM map (Fig. 1b–d and Supplementary Fig. 5). This N-terminal disordered region is responsible for the interaction with SYT1 and should therefore be located in the cytoplasm. The 12 transmembrane helices of SV2A adopt a canonical MFS fold, in which the N- and C-terminal halves, each containing six transmembrane helices, are pseudo-symmetrically related (Fig. 1b–d). A central cavity is formed between the two repeats. Two cytoplasmic horizontal helices, H1 and H3, constitute the cytoplasmic domain. The long loop extending from H3 continues to the horizontal H4 of the C-terminal half. The junction between the N- and C-halves is disordered in the cryo-EM map.

The central helices, TM1 and TM7, are spread apart to open the central cavity towards the vesicular lumen (Fig. 1c, d). LD4 is inserted between TM7 and TM8 in the C-terminal half and forms a β-helix structure (Fig. 1b–d). TM7 extends to the luminal side and is directly connected to the N-terminus of LD4. The C-terminus of LD4 and TM8 are connected by a flexible loop that is poorly defined in the cryo-EM map (Supplementary Fig. 6a). Cys583 in the LD4–TM8 loop is close to Cys198 in the TM1–TM2 loop, possibly forming a disulfide bond (Supplementary Fig. 6a). However, the quality of the cryo-EM density in this region was insufficient to define disulfide bond formation. Therefore, we performed a mass spectrometry analysis using the purified SV2A (Supplementary Fig. 6b, c). The SV2A used in this study was purified without any reducing reagent. Our mass spectrometry analysis did not detect any disulfide-crosslinked peptides (see "Methods"). For further investigation, the C198S and the C583S mutant SV2As were overexpressed and purified (Supplementary Fig. 7). Both mutant proteins showed expression levels and size-exclusion chromatography profiles comparable to the wild-type protein. Therefore, the SV2A used in this study appears not to contain a disulfide bond, although one cannot rule out the formation of a disulfide bond depending on the redox condition of SV in neuronal cells.

We also determined the cryo-EM structure of the SV2A–LEV complex without $H_C$A2 to a nominal resolution of 3.38 Å (Supplementary Figs. 8, 9). SV2A forms a dimer, in which the two protomers are in an inverted configuration in the detergent micelle (Supplementary Fig. 10). Thus, the observed SV2A dimer is functionally irrelevant. The overall topologies of the SV2As are essentially the same in the presence and absence of $H_C$A2 (Supplementary Fig. 10c). The cryo-EM density of LD4 is less resolved than that of TMD, suggesting a flexible movement of LD4 relative to TMD. This is further supported by the higher temperature B-factors of LD4 in the local resolution map (Supplementary Fig. 10d). Our MD simulations of SV2A embedded in 1-palmitoyl-2-oleoyl-glycero-3-phosphocoline (POPC) membrane also suggest a flexible movement of LD4 (Supplementary Fig. 11).

### BoNT/A2 binding site

The open edge of the β-strand from the β-hairpin of $H_C$A2 forms backbone hydrogen bonding interactions with the open edge of the C-terminal β-strand from the β-helix structure of LD4A (Fig. 2a, b). In

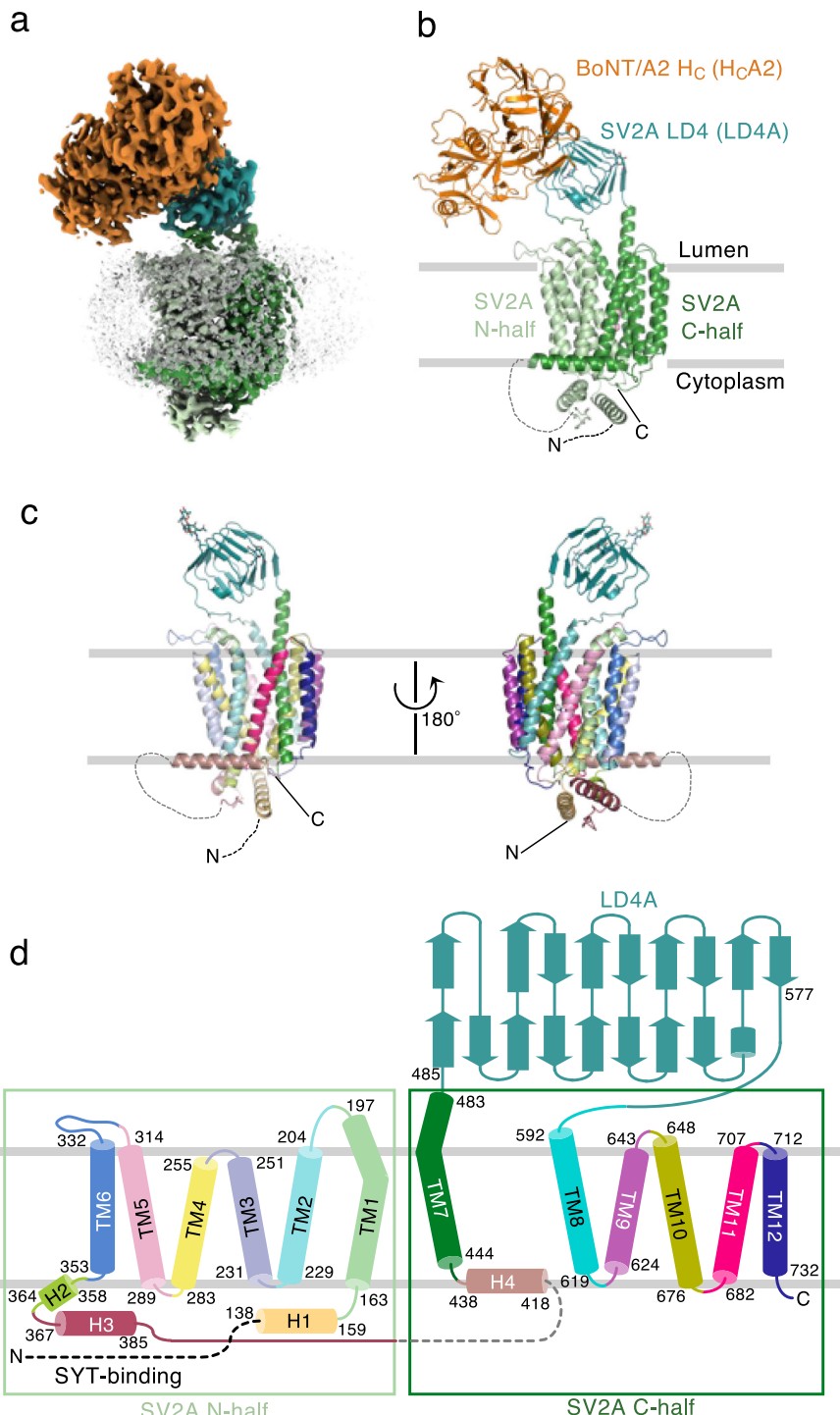

**Fig. 1 | Cryo-EM structure of the SV2A–BoNT/A2 H$_C$ complex bound to LEV.**
**a** Cryo-EM map of the SV2A–BoNT/A2 H$_C$ complex bound to LEV. The globally refined map of the SV2A–H$_C$A2–LEV complex and the local refinement map of the LD4A–H$_C$A2 complex are combined. The N- and C-terminal halves of the SV2A TMD are colored in pale and forest green, respectively. LD4A is colored in deep teal, and H$_C$A2 is colored in orange. A micelle-like density (gray) surrounds SV2A TMD.

**b** Ribbon representation of the SV2A–H$_C$A2–LEV complex. The coloring scheme is the same as that in (**a**). **c** SV2A structure in different colors for each secondary-structure element. **d** Topology diagram of SV2A. The beginning and end of each secondary-structure element are numbered. The coloring scheme is the same as that in (**c**).

addition to the backbone–backbone interaction, the SV2A His578 side chain forms a salt bridge with the BoNT/A2 Glu1156 side chain. SV2A Phe576 contributes to the van der Waals interactions with Ser1142 and Val1144 of BoNT/A2. Finally, the BoNT/A2 Tyr1149 side chain forms a hydrogen bond with the SV2A Asn573 side chain and supports the attached N-glycans for the interaction mentioned below.

The protein–protein interaction between H$_C$A2 and LD4A creates a relatively small binding interface with a buried surface area of 464 Å². To enhance this interaction, N-glycans attached to LD4A contribute to binding[45]. Three putative N-glycan sites, Asn498, Asn548, and Asn573 in SV2A, are mapped on LD4A. Masked local refinement revealed the well-ordered densities of the N-glycans at

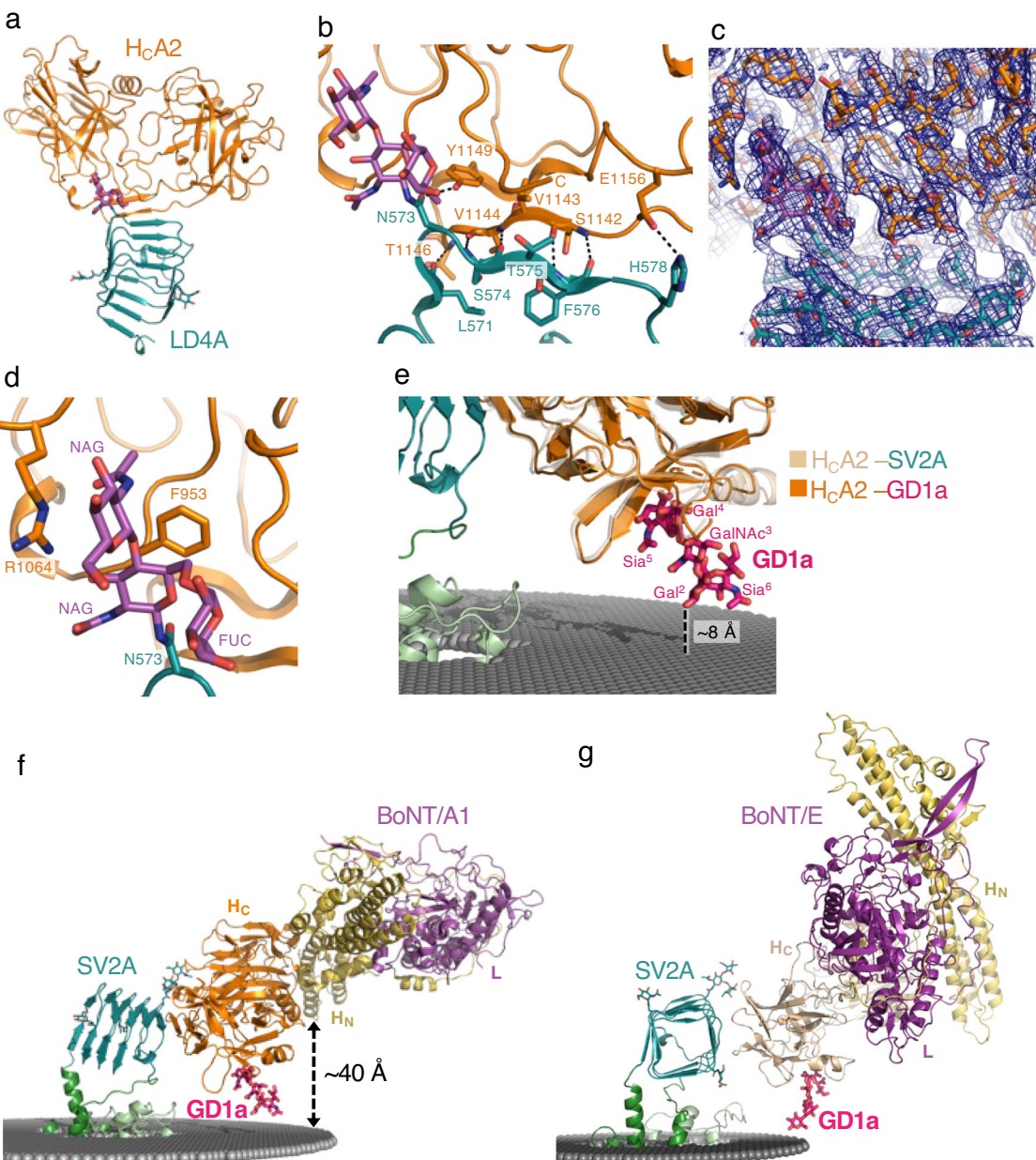

**Fig. 2 | Interaction between SV2A and BoNT. a** The structure of the LD4A–H$_C$A2 interaction. **b** The protein–protein interaction between H$_C$A2 and LD4A. The interacting residues are shown as sticks, and the hydrogen bonds are shown as dashed lines. **c** Cryo-EM map of the binding interface between LD4A and H$_C$A2. **d** The protein–glycan interaction between H$_C$A2 and LD4A. H$_C$A2 recognizes the N-glycans attached to SV2A Asn573. **e** Superposition of H$_C$A2–GD1a complex (PDBID = 7Z5S) onto the SV2A–H$_C$A2–LEV complex. **f** Model of the full-length BoNT/A1 bound to SV2A. **g** Model of the full-length BoNT/E bound to SV2A as a dual-receptor complex.

the binding interface (Fig. 2c and Supplementary Fig. 3b). A *N*-glycan containing two *N*-acetylglucosamines (NAG)s and fucose (FUC) attached to Asn573 forms extensive contacts with H$_C$A2 (Fig. 2d). In insect cells, the attached *N*-glycans are mainly paucimannose, containing with two NAG, a FUC, and three mannose[54]. By contrast, mammalian *N*-glycans are often complex type, which has an elongated structure from paucimannose with terminally galactosylated or syalylated. Thus, in either insect cells or mammalian cells, their *N*-glycans contain two NAGs and a FUC as a core essential for recognition by BoNT. The BoNT/A2 Phe953 and the aliphatic portion of the BoNT/A2 Arg1064 form the van der Waals contacts with the two NAGs attached to LD4A.

The LD4A structure in our cryo-EM structure forms a β-helix structure similar to those previously reported LD4C structures with an

rmsd of ~0.7 Å for comparable Cα atoms, despite the moderate sequence similarity (53% identical residues) between LD4A and LD4C (Supplementary Fig. 12a). When our SV2A–H$_C$A2–LEV structure is compared with the unglycosylated LD4C–H$_C$A2 structure (PDBID = 6ES1)[55], both structures share the common backbone–backbone interactions (Supplementary Fig. 12b, c). Our SV2A–H$_C$A2–LEV structure is also compared with the glycosylated LD4C–BoNT/A1 H$_C$ (H$_C$A1) structure (PDBID = 5JLV; Supplementary Fig. 12d)[53]. The notable difference is that the BoNT/A1 Arg1294 near the C-terminus lies on the β-helical bundle of LD4C to form the van der Waals contacts, whereas the corresponding residue of BoNT/A2 is Ser1294 (Supplementary Fig. 12e). In addition, Arg1064 in BoNT/A2 is replaced by His1064 in BoNT/A1, which plays a similar role in the interaction with NAG attached to LD4C.

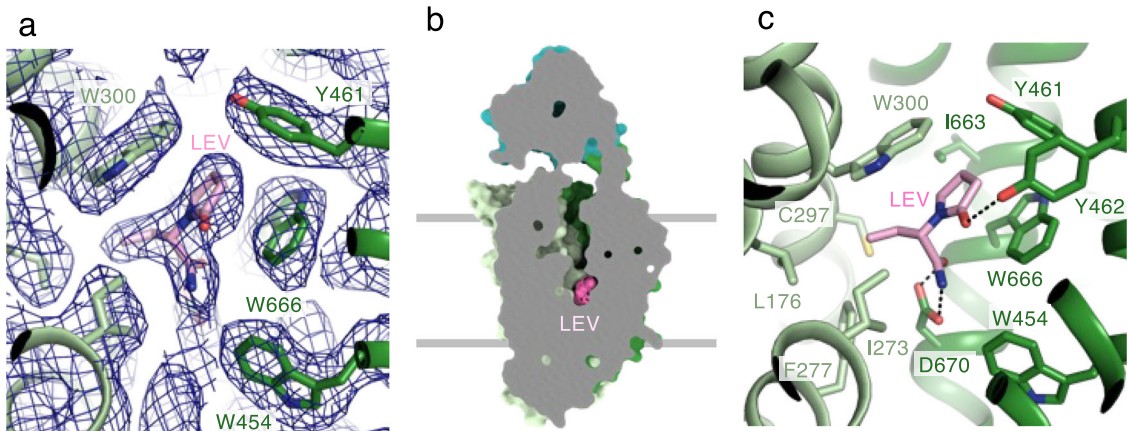

**Fig. 3 | Levetiracetam binding site. a** The cryo-EM map (blue mesh) of the LEV-binding site. **b** Cutaway surface representation of the SV2A in complex with LEV, which is shown in spheres (hot pink). SV2A is in outward-open conformation. **c** Close-up view of the LEV-binding site. The residues interacting with LEV are indicated by sticks.

Most BoNT H$_C$s can bind to gangliosides, which are rich on the neuronal cell surface, and simultaneously bind to their corresponding protein receptor, forming a dual-receptor complex[41]. The structures of the BoNT/A H$_C$s from the different subtypes in complex with ganglioside revealed that they form a common binding pocket containing a conserved SxWY motif[32]. When the H$_C$A2–GD1a structure (PDBID = 7Z5S)[56] is superposed onto our SV2A–H$_C$A2–LEV structure, GD1a is located close to the putative membrane surface (Fig. 2e). The distance between GD1a Gal2 and the membrane surface is ~8 Å in good agreement with the presence of a glucose residue between Gal2 and a hydrophobic ceramide tail, supporting the dual-receptor mechanism of BoNT/A. Superposition of the full-length BoNT/A1 structure (PDBID = 2NYY)[57] onto the SV2A–H$_C$A2–LEV structure reveals that two long helices of the translocation domain (H$_N$) are positioned almost parallel to and about 40 Å away from the membrane surface (Fig. 2f). Since two long helices of the translocation would play a crucial role in membrane insertion, the translocation domain must undergo a conformational rearrangement presumably induced by low pH in SV.

The recently reported structure of the H$_C$E–LD4AC complex displayed an unexpected binding mode, in which H$_C$E binds to the side of the β-helical bundle of LD4 (Supplementary Fig. 12f). We superposed the H$_C$E–LD4AC structure (PDBID = 7UIB)[52] onto the SV2A–H$_C$A2–LEV structure to model the SV2A–H$_C$E structure. Then, the full-length BoNT/E structure (PDBID = 3FFZ)[58], which forms a "closed butterfly" conformation, together with H$_C$E–GD1a structure (PDBID = 7OVW)[59], was superposed onto the SV2A–H$_C$E model to build the complete BoNT/E–SV2A model. However, GD1a is located ~24 Å apart from the membrane surface in this model. Given a certain flexible movement of LD4A, we manually repositioned LD4A and reasonably modeled the complete BoNT/E–SV2A structure as a dual-receptor complex (Fig. 2g).

### Levetiracetam binding site
An unambiguous density, assigned as LEV, was observed at the base of the central cavity (Fig. 3a, b). Substrate-binding sites are typically located at the base of the central cavity of MFS transporters. Therefore, LEV is likely to occupy the binding pocket of an unidentified SV2A substrate. A narrow pathway is formed from bound LEV to the extracellular space (Supplementary Fig. 13), indicating that SV2A adopts an outward-open conformation (intravesicular-facing conformation). We also determined the cryo-EM structure of the SV2A–H$_C$A2 complex without LEV at 3.05 Å resolution (Supplementary Figs. 14, 15). Although an unidentified density was observed in the LEV-binding site, the size and shape of this density were apparently different from those of LEV observed in the SV2A–H$_C$A2–LEV complex (Supplementary Fig. 16a). The apo SV2A structure is virtually indistinguishable from

that bound to LEV, with an rmsd of 0.3 Å (Supplementary Fig. 16b), suggesting that the outward-open conformation of SV2A is pre-organized to engage LEV.

The aromatic residues (Phe277, Trp300, Trp454, Tyr461, Tyr462, and Trp666) constitute the LEV-binding pocket (Fig. 3c). The γ-lactam ring of LEV creates a π–π stacking interaction with Trp300 on TM5 and Trp666 on TM10. The γ-lactam ring is further surrounded by hydrophobic residues, such as Tyr461, Tyr462, and Ile663. In addition, the O1 of the γ-lactam ring forms a hydrogen bond with the hydroxyl of Tyr462. The carboxyl and amide groups of the butanamide moiety in LEV form hydrogen bonds with the Asp670 side chain. Leu176, Ile273, Phe277, and Cys297 form van der Waals interactions with the butanamide ethyl group in LEV. The residues involved in the LEV-binding are perfectly conserved in SV2B and SV2C with two exceptions that Ile273 and Cys297 in SV2A are substituted with Leu and Gly in SV2B, respectively (Supplementary Fig. 5). These two substitutions may reduce the binding ability of SV2B to LEV. However, the highly conserved LEV-binding site in SV2 proteins is inconsistent with the strict specificity of LEV to SV2A, suggesting an unknown modulation mechanism in SV2B and SV2C[2].

Previous extensive mutagenesis studies have identified the residues involved in the interaction with racetam derivatives[60–62]. Most of these identified residues are mapped near the LEV-binding site in our cryo-EM structure (Supplementary Fig. 17). The I273A, F277A, W300A, I663A, W666A, and D670A mutants show significantly reduced binding to the racetam groups[60,61]. Alkylation of Cys297 abolished binding, presumably by blocking the binding site[60]. In addition, Trp454, Lys694, Val661, and Asn667, whose mutations are also reported to reduce the binding ability, are close to the LEV-binding site[60–62].

### Brivaracetam binding site
We further determined the cryo-EM structure of the SV2A–H$_C$A2 complex bound to BRV at 3.11 Å resolution (Supplementary Figs. 18, 19). SV2A adopts an outward-open conformation that is virtually identical to that in complex with H$_C$A2 and LEV. BRV occupies the same binding site as LEV (Fig. 4a). A propyl group attached with γ-lactam ring in BRV forms van der Waals contacts with Tyr461 and Ile663, explaining the higher affinity of BRV for SV2A than that for LEV.

To assess the importance of Tyr461 and Ile663 in the binding of SV2A to BRV, a quantitative analysis of the interaction is required. This is particularly difficult due to the mostly negligible changes in size and conformation of SV2A upon binding to BRV. We found that the label-free spectral shift of the SV2A intrinsic tryptophan fluorescence at 350 nm and 330 nm changes significantly upon binding to BRV (Supplementary Fig. 20)[63]. Two tryptophan residues, Trp300 and Trp666,

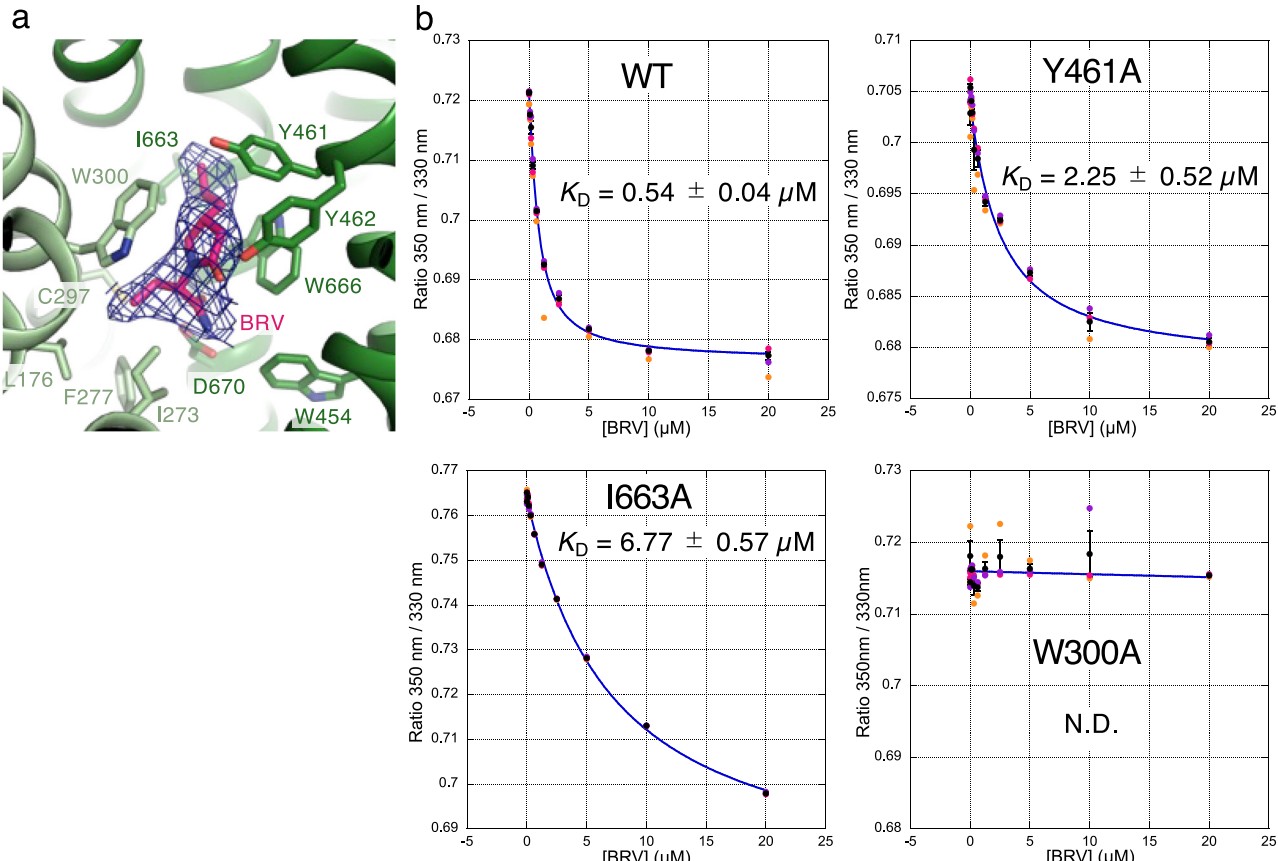

**Fig. 4 | Brivaracetam binding site. a** Close-up view of the BRV binding site. The residues interacting with BRV are indicated by sticks. The cryo-EM density of the bound BRV is shown in blue mesh. **b** Label-free spectral shift assay. Ratios of the SV2A intrinsic tryptophan fluorescence at 350 nm/330 nm with a given concentration of BRV are plotted. Prots (black) show the mean derived from $n = 3$ technical replicates (red, orange, and purple circles) and the error bars show the standard deviations. Source data are provided as a Source Data file.

which form a π–π stacking interaction with the γ-lactam ring of BRV, must contribute to this label-free spectral shift, as their local environments differ significantly upon binding to BRV. From the dose-response curve of the spectral shift, the equilibrium dissociation constant ($K_D$) of the interaction between the wild-type SV2A and BRV was calculated to be ~540 nM (Fig. 4b), which is comparable to the previously reported $K_D$ values using the radio-labeled BRV[4,61]. The Y461A and the I663A mutant SV2As reduced their binding affinities by ~4-fold and ~12-fold, respectively, supporting the structure. As noted above, Trp300 is the most critical residue for interaction with LEV and BRV, and its replacement to alanine abolished binding to racetam derivatives in the previous report[60]. Accordingly, the W300A mutant SV2A did not show a dose-dependent response in our spectral shift measurement.

## Discussion

SV2 proteins are relatively abundant proteins in SV[64] and were identified nearly 40 years ago as the antigen for the monoclonal antibody raised against the SVs from the electric ray *Discopyge ommata*[65]. Despite extensive studies, the exact functions of these proteins remain unclear. SV2s were initially thought to be neurotransmitter transporters. However, SV2A is present in almost all types of neurons, ruling out the possibility that it transports a specific type of neurotransmitter[7,8]. In addition, the vascular neurotransmitter transporters for each neurotransmitter have been identified[66]. Heterologous expression of SV2A in hexose transporter-deficient *Saccharomyces cerevisiae* restores growth in galactose-containing medium, demonstrating that SV2A is capable of transporting the extracellular galactose into the cell[67]. LEV

inhibits the galactose-dependent growth of yeast cells expressing SV2A. We tested the galactose-binding of SV2A using our label-free spectral shift assay. However, no obvious spectral shift was detected (Supplementary Fig. 21a, b), though the LEV-binding pocket of SV2A appears to be large enough to accommodate galactose in our manual docking model (Supplementary Fig. 21c). Thus, it remains elusive if SV2A directly transport galactose. Mutating of Trp300 and Trp666, the most critical residues for binding to LEV and BRV, failed to rescue synaptic transmission in neurons from SV2A/B DKO mice[68]. These mutations did not affect SV2 trafficking to the synapse, SYT1 expression, and SYT1 internalization[68], and therefore the impairment of synaptic transmission by these mutations suggests an essential role of SV2A as a membrane transporter.

If SV2A functions as a membrane transporter, it might undergo the outward-open to inward-open conformational transition of the canonical alternate access model, possibly driven by the low internal pH of SV. The previous in situ visualization of SV2A using the protein tomography technique detected the intravesicular-facing and the cytoplasmic-facing conformations[69]. To gain an insight into the proton-driven conformational change of SV2A, we mapped Asp, Glu, His, Arg, and Lys residues in TMD (Supplementary Fig. 22a). There are only three acidic residues inside the cavity; one is Asp670 involved in the direct binding to LEV and BRV, and the others are Asp179 and Glu182 on TM1 (Supplementary Fig. 22b). The Asp179 side chain points to TM4, which is slightly distorted at the region encompassing $G^{268}$-$I^{269}$-$G^{270}$-$G^{271}$-$S/A^{272}$-$I/L^{273}$-$P^{274}$, suggesting its intrinsic flexibility (Supplementary Figs. 22c, 5). Asp179 appears to stabilize the intrinsic flexible region of TM4 through a hydrogen bond with the main chain O atom of

Ile269. The Glu182 side chain forms a hydrogen bond with Arg262 (Supplementary Fig. 22b). TM1 and TM7 are spread apart to open the central cavity towards the vesicular lumen. The conformation of the vesicular lumen region of TM1 is stabilized by hydrogen bonds made through Glu194, Asp196, Lys204, His331, and Arg334 (Supplementary Fig. 22d).

We hypothesized the proton-driven conformational transition model triggered by the local rearrangement of TM1 and TM4 (Supplementary Fig. 22e). The protonated Glu182 on TM1 at low internal pH of SV leads to the repulsion away from Arg262 on TM4. The low internal pH may also break the hydrogen bond network of the vesicular lumen region of TM1. The resulting increased flexibility of TM1 breaks the hydrogen bond between Asp179 and Ile269, destabilizing the potentially flexible region of TM4. Taken together, TM1 moves away from TM4 and towards TM7 to close the extracellular entrance of the cavity. TM4 undergoes the conformational rearrangement to open the cytoplasmic side. These two helices' movements might induce the rotation of the N-terminal half relative to the C-terminal half for the inward-open conformation. The single mutations of Asp179 and Glu182 (the D179A and the E182A mutant proteins) are reported not to affect the binding to racetam derivatives[60]. The double mutation of these acidic residues (the D179A/E182A or the D179N/E182Q mutant proteins) disrupts the synaptic localization of SV2A[29,68]. These suggest that the interhelical hydrogen bonds (Asp179–Ile269 and Glu182–Arg262) play an important role in protein folding, and loss of these hydrogen bonds, triggered by protonation at Glu182, may induce the conformational transition for transport.

Our cryo-EM structures also provide detailed information on the interaction between intact SV2A and the BoNT/A2 $H_C$ domain. Intact SV2A allows us to elucidate the precise positioning of BoNT/A relative to the membrane, supporting the dual-receptor mechanism of BoNT/A[41]. Two distinct protein receptors of BoNTs, SV2A, and SYT1, colocalize with ganglioside to form tripartite nanoclusters on the neuronal plasma cell membrane, which are internalized into SV by the endocytic machinery[70,71]. Live cell super-resolution microscopy demonstrated the targeting of BoNT/A to the tripartite ganglioside–SYT1–SV2 nanoclusters on the neuronal plasma membrane[72]. SYT1 knock-down results in loss of BoNT/A toxic function[72], although BoNT/A utilizes SV2A as its protein receptor, suggesting that the tripartite nanocluster formation is essential for endocytic targeting of BoNT/A. The different BoNT serotypes, utilizing either SYT1 or SV2A as their protein receptor, may target this tripartite nanocluster and share a common endocytic entry mechanism.

It has been suggested that the highly glycosylated SV2A might provide an internal gel matrix, which might facilitate the retention of neurotransmitters inside SV[73]. As mentioned before, the LD4 of SV2A is glycosylated. The cryo-EM structure of the dimeric assembly of the SV2A–$H_C$A2–LEV complex is formed through the LD4–LD4 interaction, in which the open edge of the β-strand from each LD4 forms an antiparallel β-sheet (Supplementary Fig. 4b, c). The association of LD4 may contribute to accumulating the glycans to form a gel matrix. The TMDs in the dimeric complex are in near perpendicular orientation (Supplementary Fig. 4a, b). Together with a certain flexibility of LD4 relative to TMDs, it could be postulated that the dimeric complex might accommodate the membrane curvature of SV and stabilize or regulate the size and morphology of the vesicle. However, the previous electron microscopy analysis reported that the size, number, and location of SV appeared to be unchanged in the neurons from either SV2A/B DKO or SV2A KO mice[18,19]. Further study is necessary to address the structural role of SV2A in SV morphology.

Three pathogenic mutations of SV2A have been reported: R383Q, R570C, and G660R (Supplementary Fig. 23a). A homozygous R383Q mutation was found in a patient with intractable epilepsy, involuntary movements, microcephaly, and developmental and growth retardation[15]. The R383Q mutation caused mislocalization of

SV2A to the plasma membrane in cultured mouse hippocampal neurons[74]. Arg383 is located on the horizontal H3 in the cytoplasm and is likely to interact with phospholipids together with the neighboring positively charged residues, Arg163, Lys375, Lys385, and Arg390, (Supplementary Fig. 23b), potentially contributing to the correct localization of SV2A. A heterozygous R570C mutation was found in a patient and mother with epilepsy and poor response to LEV[16]. Arg570 is located on the surface of LD4A and forms a hydrogen bond with Asn548, which is an N-glycan attachment site (Supplementary Fig. 23c). The R570C mutation would interfere with the proper folding and trafficking of SV2A. Finally, a heterozygous G660R mutation was found in a patient with new-onset epilepsy, who had another seizure after treatment with LEV[17]. Gly660 is located on TM10 and is closest to the neighboring TM8 (Supplementary Fig. 23d). Its mutation to a bulky arginine residue may disrupt the interhelical packing and destabilize the overall structure.

In summary, our structural studies unveil the racetam-based AEDs and BoNT recognition by SV2A. Recent genetic analyses identified pathogenic mutations in SV2A, suggesting an essential role of SV2A for normal synaptic transmission. Although the exact function of SV2A remains unclear, our study suggests its potential role as a membrane transporter. LEV and BRV may function as inhibitors of the SV2A transporter function. Our cryo-EM structure provides a structural framework to elucidate the binding mode of the preexisting racetam derivatives of the AEDs and PET tracers and to rationally design additional derivatives.

## Methods
### Protein expression and purification
The gene encoding *human sv2a* (UniprotID = Q7L0J3: residues 2–742) was PCR-amplified and cloned into the pLIB vector[75]. SV2A fused with the N-terminal FLAG tag was transformed into *Escherichia coli* DH10multibac competent cells to generate the baculovirus DNA using a bac-to-bac system (Thermo Fisher Scientific). Baculovirus bearing *sv2a* was generated in *Spodoptera frugiperda* Sf9 insect cells. The Sf9 cells were transfected with the baculoviruses bearing *sv2a* and incubated at 27 °C for 72 h. The cell pellets were resuspended in a lysis buffer containing 20 mM Tris-HCl (pH 8.0), 300 mM NaCl, and 10% glycerol, and homogenized using a Dounce homogenizer. After ultracentrifugation at 186,000×*g* at 4 °C for 40 min, the membrane pellet was solubilized in a lysis buffer containing 1% LMNG and 0.2% cholesterol hemisuccinate (CHS) at 4 °C for 2 h. After centrifugation at 186,000×*g* for 40 min, the supernatant was collected and applied to anti-FLAG M2 (Sigma) resin. The resin was washed with 10 column volumes of buffer containing 20 mM Tris-HCl (pH 8.0), 300 mM NaCl, 10% glycerol, 0.001% LMNG, and 0.0002% CHS. The bound protein was eluted with the FLAG peptide in 20 mM Tris-HCl (pH 8.0), 300 mM NaCl, 10% glycerol, 0.001% LMNG, and 0.0002% CHS. The protein was then further purified by size-exclusion chromatography using a Superdex 200 10/300 GL (Cytiva) preequilibrated with the SEC buffer containing 20 mM HEPES (pH 7.5), 150 mM NaCl, 0.001% LMNG, and 0.0002% CHS. The SEC profile showed two peaks corresponding to the dimeric and monomeric SV2A (Supplementary Fig. 1a). Peak fractions containing the monomeric SV2A were pooled and concentrated using a 100 kDa cutoff concentrator (Amicon). All the mutant proteins used in this study were purified in the same manner as the wild-type protein. Uncropped gel images are shown in Supplementary Fig. 25.

To prepare the SV2A dimer without $H_C$A2, SV2A was coexpressed with SYT1, as we initially intended to copurify them. The protein was extracted from the membrane and purified with immunoaffinity purification using anti-FLAG M2 resin in the same manner described above. However, SYT1 was barely detectable owing to its negligible expression. SV2A alone was further purified by SEC using a Superdex 200 10/300 GL (Cytiva) preequilibrated with the SEC buffer containing 20 mM HEPES (pH 7.5), 150 mM NaCl, and 0.004% GDN. The peak

fraction containing the dimeric SV2A was collected and concentrated (Supplementary Fig. 1b).

The gene encoding BoNT/A2 (UniprotID = Q45894) Hc was codon-optimized, synthesized (Eurofins genomics), and cloned into the pCold-vector (Takara Bio). $H_C$A2 fused with an N-terminal 6×His-SUMO tag[76] was overexpressed in BL21 Gold (DE3) *E. coli* cells (Agilent technology), and the cells were disrupted using sonication. After centrifugation, the clarified supernatant was applied to the HisTrap column (Cytiva) preequilibrated with a buffer containing 20 mM imidazole, 20 mM Tris-HCl (pH 8.0), and 500 mM NaCl. The resin was washed with 10 column volumes of the same buffer. The bound proteins were eluted with a buffer containing 500 mM imidazole, 20 mM Tris-HCl (pH 8.0), and 500 mM NaCl. The N-terminal 6×His-SUMO tag was cleaved off using a homemade ULP1 protease[76] during dialysis against a buffer containing 20 mM imidazole, 20 mM Tris-HCl (pH 8.0), and 300 mM NaCl. $H_C$A2 was reapplied onto the HisTrap column, and the flowthrough fractions were collected. $H_C$A2 was further purified by SEC using a HiLoad 16/60 Superdex 200 (Cytiva) preequilibrated with the SEC buffer containing 20 mM HEPES (pH 7.5) and 150 mM NaCl (Supplementary Fig. 1c).

## Cryo-EM sample preparation and data collection

Purified SV2A and $H_C$A2 were mixed at a molar ratio of 1:1.5 to form the SV2A–$H_C$A2 complex. Then, LEV was added at a final concentration of 0.1 mM for the SV2A–$H_C$A2–LEV complex. For the SV2A–LEV dimer, the SV2A dimer purified in GDN was mixed with LEV. BRV was prepared from Brivaracetam-D3 solution in methanol (Supleco) by removing methanol by evaporation and dissolved in a buffer containing 20 mM HEPES (pH 7.5) and 150 mM NaCl. BRV was added at a final concentration of ~90 μM to the SV2A–$H_C$A2 complex. A 3 μL sample was applied to a glow-discharged holey carbon grid (Quantifoil Au or Cu, 300 mesh, R1.2/1.3). The grid was blotted for 3 s and plunged into liquid ethane under 100% humidity at 4 °C using a Vitrobot Mark IV (Thermo Fisher Scientific).

Micrographs were acquired on a Titan Krios microscope (Thermo Fisher) operating at 300 kV equipped with a K3 direct electron detector (Gatan) and a Bioquantum energy filter with a slit width of 15 eV at a magnification of 105,000×, resulting in a pixel size of 0.83 Å, using EPU software (Thermo Fisher Scientific). A total of 6518 movie frames of the SV2A–$H_C$A2–LEV complex were captured using a total dose of 50.8 e⁻/Å² for 48 frames with an exposure time of 2.1 s. A total of 10,001 movie frames for the SV2A–LEV dimer were captured using a total dose of 50.2 e⁻/Å² for 48 frames with an exposure time of 2.1 s. A total of 4890 movie frames of the SV2A–$H_C$A2 complex was captured using a total dose of 50.0 e⁻/Å² for 48 frames with an exposure time of 2.1 s. A total of 6001 movie frames of the SV2A–$H_C$A2–BRV complex was captured using a total dose of 53.5 e⁻/Å² for 48 frames with an exposure time of 2.0 s. Collected movie frames were imported into cryo-SPARC-4.2.1[77] and processed through patch motion correction and patch CTF estimation.

For the SV2A–$H_C$A2–LEV complex, initial particle picking was performed with a reference-free blob picker, followed by particle extraction with a box of 256 pixels and binned to 64 pixels. After a reference-free 2D classification, the selected 2D classes with distinguishable TMD were used as templates for template-based picking, resulting in a total of 6,409,122 particles. After 2D classification, 1,622,608 particles were selected and subjected to an ab initio reconstruction. Heterogeneous refinement was performed using four 3D references. Two distinct good classes with the distinguishable TMD of SV2A were selected: the monomeric and the 2:2 assembly of the SV2A–$H_C$A2–LEV complex. For the monomeric complex, the selected particles (660,976) were re-extracted with a box of 300 pixels and binned to 150 pixels. After homogeneous refinement, the particles were re-extracted with a box of 300 pixels, followed by a single round of the homogeneous refinement. Then, the non-uniform (NU) refinement was performed[78], resulting in a 2.88 Å resolution map. The quality of the cryo-EM map was sufficient for

the de novo building of the SV2A in complex with $H_C$A2 and LEV. Local refinement using the mask covering $H_C$A2–LD4A was performed to improve the peripheral region of the bound $H_C$A2, resulting in a 2.82 Å resolution map. For the 2:2 complex, the selected particles (555,697) were subjected to another round of heterogeneous refinement. The selected particles were re-extracted with a box of 300 pixels and binned to 150 pixels. After homogeneous refinement, the particles (425,512) were re-extracted with a box of 360 pixels and subjected to NU-refinement. The cryo-EM map of the 2:2 complex was reconstructed to a nominal resolution of 2.90 Å, showing that two of the SV2A–$H_C$A2–LEV complexes are assembled through the LD4–LD4 interactions.

For the SV2A–LEV dimer, particle picking was performed with a reference-free blob picker, followed by template-based picking with a box of 256 pixels and binned to 64 pixels. In total, 6,435,573 particles were selected and subjected to 2D classification. Subsequently, 1,949,069 particles were selected and subjected to an ab initio reconstruction. Heterogeneous refinement was performed using four 3D references. One class was selected, corresponding to the dimeric particles, in which the two SV2A protomers are in the inverted configuration. The selected particles (829,253) were re-extracted with a box of 360 pixels and binned to 180 pixels. After the ab initio reconstruction, heterogeneous refinement was performed using three 3D references. Again, one class was selected, followed by a single round of NU-refinement. The particles (434,198) were re-extracted with a box of 360 pixels and subjected to NU-refinement. The masked local refinement resulted in a 3.38 Å resolution map.

For the SV2A–$H_C$A2 complex (without LEV), particle picking was performed with a reference-free blob picker, followed by template-based picking, resulting in 5,101,376 particles. After 2D classification, 1,146,283 particles were selected and subjected to an ab initio reconstruction. Heterogeneous refinement was performed using four 3D references, and a good class was selected. The selected particles (604,695) were subjected to 3D classification using principal component analysis (three classes). Two classes of 356,711 particles were subjected to an ab initio reconstruction. Homogeneous refinement was performed using a new 3D reference, and the particles were re-extracted with a box of 300 pixels and binned to 150 pixels. After homogeneous refinement, the particles were re-extracted with a box of 300 pixels, followed by a single round of homogeneous refinement. Then, NU-refinement was performed, resulting in a 3.01 Å resolution map. The masked local refinement using the mask covering $H_C$A2–LD4A was performed, resulting in a 2.87 Å resolution map.

For the SV2A–$H_C$A2–BRV complex, particle picking was performed with a reference-free blob picker, followed by template-based picking, resulting in 4,734,449 particles. After 2D classification, 1,222,077 particles were selected and subjected to an ab initio reconstruction. Heterogeneous refinement was performed using four 3D references, and a good class (531,619 particles) was selected. The particles were re-extracted with a box of 300 pixels and binned to 150 pixels, followed by 2D classification. The selected particles (487,356) were used for NU-refinement. The particles (479,130) were re-extracted with a box of 300 pixels and subjected to NU-refinement, resulting in a 3.11 Å resolution map. The masked local refinement using the mask covering $H_C$A2–LD4A was performed, resulting in a 3.07 Å resolution map.

## Model building and refinement

For the model building of the SV2A–$H_C$A2 complex bound to LEV, we first built the atomic model of the $H_C$A2–LD4A complex from the 2.82 Å resolution map of the local refinement covering $H_C$A2 and LD4A. The $H_C$A2–LD4C complex structure (PDBID = 6ES1) was used as an initial model. The atomic model was then manually adjusted using Coot-0.9.8.1[79,80]. The atomic model of the SV2A–$H_C$A2–LEV complex was first built from the globally sharpened 2.88 Å resolution map. The $H_C$A2–LD4A structure was used as the initial model for $H_C$A2 and LD4A. The TMD of SV2A was manually built using Coot-0.9.8.1. The LEV

 

structure (The Cambridge Structural Database: 1217172) was used as the initial model for bound LEV. The ligand restraint files were generated using eLBow in Phenix-1.19.2[81,82]. The model was manually adjusted using Coot-0.9.8.1. The structure was refined using real-space refinement in Phenix-1.19.2[83] and further refined against unweighted half maps using Refmac Servalcat[84] in CCPEM-1.6.0[80]. The final model was built from the composite map and refined using Phenix-1.20.1[83]. The 2:2 assembly of the SV2A–$H_C$A2–LEV complex was modeled using the monomeric SV2A–$H_C$A2–LEV complex. The final model contains the SV2A–$H_C$A2–LEV complex and the LD4A–$H_C$A2 complex, as the TMD of the second protomer showed substantially weaker densities. The final model was refined using Refmac Servalcat[84].

The SV2A–LEV dimer was built from the globally sharpened 3.38 Å resolution map, using the SV2A–LEV monomeric complex from the SV2A–$H_C$A2–LEV complex structure as the initial model. The model was refined using Refmac Servalcat[84]. For the SV2A–$H_C$A2 complex, the model was built and refined similarly to that for the SV2A–$H_C$A2–LEV complex. We first built the atomic model of the $H_C$A2–LD4A complex from the local refinement map. Then, the final model of the SV2A–$H_C$A2 complex was built from the composite map and refined using Phenix-1.20.1[83]. The model of the SV2A–$H_C$A2–BRV complex was built from the composite map of the global refinement map and the local refinement map. The model of the BRV was generated by adding a propyl group to the C4 of the γ-lactam ring of LEV using UCSF Chimera[85]. The final model of the SV2A–$H_C$A2–BRV complex was refined using Phenix-1.20.1[83]. The stereo chemistries for the final models were assessed using MolProbity[86] and summarized in Supplementary Table 1. The buried surface area was calculated using PISA-2.1.0[87]. The lipid bilayer was predicted using the Positioning of Proteins in Membrane (PPM) server[88]. The transport pathway in the SV2A was calculated using Caver-3.0.3[89]. All figures were prepared using PyMol-2.5.2 (https://pymol.org) and UCSF Chimera X-1.2.5[90].

## Mass spectrometry
The purified SV2A solution was denatured by adding 7 M guanidine-HCl/1 M Tris (pH 8.5)/10 mM EDTA, and free sulfhydryl groups were modified by adding iodoacetate. The samples were desalted using a PAGE Clean Up Kit (Nacalai Tesque). The precipitated proteins were digested with trypsin (TPCK-treated, Worthington Biochemical Corporation) and endoproteinase Glu-C (Roche). The resulting peptides were analyzed using a Q Exactive Hybrid Quadrupole-Orbitrap mass spectrometer (Thermo Fisher Scientific) coupled to an Easy nLC1000 (Thermo Fisher Scientific). Peptides were separated using a reversed-phase nano-spray column (NTCC-360/75-3-105, NIKKYO Technos). The mass spectrometer was operated in positive mode, and data were obtained using the TOP10 method. The acquired data were analyzed using MASCOT (version 2.8, Matrix Science) and Proteome Discoverer (version 3.0, Thermo Fisher Scientific). Mascot searches were performed using the following parameters: database = in-house database; enzyme = trypsin; maximum missed cleavages = 3; variable modifications = Acetyl (Protein N-term), Gln -> pyro-Glu (N-term Q), Oxidation (M), Carboxymethyl (C); product mass tolerance = ± 15 ppm; product mass tolerance = ± 30 milli mass unit; instrument type = ESI-TRAP. MASCOT crosslinking analysis was also carried out to identify the disulfide-crosslinked peptides. Experiments were repeated twice independently with similar results.

## Spectral shift measurement utilizing nanoDSF instrument
Label-free spectral shift measurements were performed using the Tycho NT.6 nanoDSF instrument (Nano-Temper Technologies GmbH). Dried BRV was dissolved in the assay buffer containing 20 mM HEPES (pH 7.5), 150 mM NaCl, 0.001% LMNG, and 0.0002% CHS, to a concentration of 928 μM. SV2A was prediluted to 1 μM in the assay buffer. The 9-point dilution series was prepared by twofold serial dilution starting in the assay buffer starting from a concentration of 40 μM. The

SV2A solution was mixed with an equal volume of the serial dilution series of BRV. The final concentration of SV2A, except for the W300A mutant SV2A, in the reaction mixture was 0.5 μM, and the highest concentration of BRV was 20 μM. The final concentration of the W300A mutant SV2A in the reaction mixture was 0.24 μM, because of the slightly lower expression level of the W300A mutant SV2A (Supplementary Fig. 23). The reaction mixtures were transferred into the capillaries (cat# TY-C001; Nano-Temper Technologies GmbH). The ratios of the intrinsic tryptophan fluorescence intensities at 350 and 330 nm were measured using the Tycho NT.6 instrument (Nano-Temper Technologies GmbH). Ratio changes observed in nanoDSF are indicative of the local environment change around the tryptophan residues of the target protein upon binding to the ligand. To elucidate this, a dose-dependent curve was derived by measuring a constant concentration of the target protein against a varying concentration of the ligand. The temperature was increased from 35 to 95 °C for thermal shift assay (Supplementary Fig. 20). The initial ratios (35.2 °C) at given BRV concentrations were plotted to obtain the dose-dependent curve.

The equilibrium dissociation constants ($K_D$) were calculated from the fitting of the dose-response curve of the fluorescence ratio with a 1:1 stoichiometric binding model. The fraction ($a$) of the SV2A–BRV complex at a given BRV concentration is given as

$$a = \frac{[\text{SV2A}]_{\text{total}} + [\text{BRV}] + K_D - \sqrt{([\text{SV2A}]_{\text{total}} + [\text{BRV}] + K_D)^2 - 4 \cdot [\text{SV2A}]_{\text{total}} \cdot [\text{BRV}]}}{2 \cdot [\text{SV2A}]_{\text{total}}}$$

(1)

The dose-response curve of the initial ratio at 350/330 nm is estimated as

$$R = a \cdot R_{\text{bound}} + (1 - a) \cdot R_{\text{unbound}}$$

(2)

where R is the 350/330 nm ratio at a given BRV concentration; $R_{\text{bound}}$ is the ratio value of the SV2A–BRV complex; $R_{\text{unbound}}$ is the ratio value of SV2A alone. The curve fitting was performed using Kaleida-graph 5 Mac (Synergy Software). The experiments were repeated three times with similar results. All mutants examined by spectral shift measurements were confirmed to behave similarly to the wild-type protein in the size-exclusion chromatography, except that the W300A mutant protein showed a higher ratio of the aggregation peak (Supplementary Fig. 24). All the purified proteins showed similar thermal stability form their thermal shift curves (Supplementary Fig. 20). To test the binding to galactose, the label-free spectral shift assay was performed in the same manner except that the highest concentration of the galactose in the reaction mixture was 40 mM.

## MD simulation
MD simulations were performed using the SV2A–LEV complex from the cryo-EM structure of the SV2A–$H_C$A2–LEV complex. The structure of the missing region in the cryo-EM structure of SV2A (residues 404–417) was modeled by using Modeller 9.13[91]. The two aspartic acid residues on the surface of the ligand binding pocket (Asp179 and Asp670) were protonated, and all the histidine residues were protonated on the Nε2 atom. Since the structure of TMD of SV2A was close to that of XylE (PDBID: 6N3I)[92] with the TM-score[93] of 0.89, its coordinates were obtained from the Orientations of Proteins in Membranes (OPM) database[88] and the coordinates of the TMD of SV2A was aligned to them to calculate the positioning of SV2A in a lipid bilayer. The transformed coordinates of the SV2A–LEV complex were then embedded in a solvated lipid bilayer using the "Membrane Builder"[94] function of the CHARMM-GUI server[95]. The system was composed of a protein chain, a BRV and 220 1-palmitoyl-2-oleoyl-*sn*-glycero-3-phosphocholine (POPC) molecules, 85 $K^+$ and 84 $Cl^-$ ions, and 31,039 water molecules. The size of the initial system was 9.6 nm × 9.6 nm × 15.4 nm. The topology and the parameters of BRV were generated using the

"Ligand Reader & Modeler" function[96] of the CHARMM-GUI server and the CHARMM general force field (CGenFF)[97]. The TIP3P model[98] was used for the water molecules. The CHARMM36m force field[99] was used for the protein chain, and the CHARMM36 force field[100] was used for the other molecules. The system was energy-minimized and equilibrated in the NPT ensemble at 303.15 K and $1.0 \times 10^5$ Pa, as described previously[101]. During the equilibration, distance restraints were imposed on the atom pairs involved in the intermolecular hydrogen bonds. Finally, a 1-μs MD simulation was performed without restraints. All the MD simulations were performed using GROMACS 2022[102], with coordinates recorded every 10 ps.

### Reporting summary

Further information on research design is available in the Nature Portfolio Reporting Summary linked to this article.

### Data availability

The cryo-EM maps have been deposited in the Electron Microscopy Data Bank under accession numbers EMD-36392 (SV2A–H$_C$A2–LEV), EMD-36394 (local refinement map of EMD-36392), EMD-36616 (composite map of EMD-36392 and EMD-36394), EMD-36397 (SV2A–H$_C$A2–LEV (dimeric complex)), EMD-36398 (SV2A–LEV), EMD-36395 (SV2A–H$_C$A2), EMD-36396 (local refinement map of EMD-36395), EMD-36617 (composite map of EMD-36395 and EMD-36396), EMD-36933 (SV2A–H$_C$A2–BRV), EMD-36934 (local refinement map of EMD-36933), EMD-36935 (composite map of EMD-36933 and EMD-36934). The coordinates have been deposited in the RCSB Protein Data Bank (PDB) accession codes 8JLC (SV2A–H$_C$A2–LEV for EMD-36392), 8JLE (LD4A–H$_C$A2 for EMD-36394), 8JS8 (SV2A–H$_C$A2–LEV for EMD-36616), 8JLH (SV2A–H$_C$A2–LEV (dimeric complex) for EMD-36397), 8JLI (SV2A–LEV for EMD-36398), 8JLF (SV2A–H$_C$A2 for EMD-36395), 8JLG (LD4A–H$_C$A2 for EMD-36396), 8JS9 (SV2A–H$_C$A2 for EMD-36617), 8K77 (SV2A–H$_C$A2–BRV for EMD-36935), respectively. The mass spectrometry proteomics data have been deposited to the ProteomeXchange Consortium via the PRIDE [1] partner repository with the dataset identifier PXD050355 [doi.org/10.6019/PXD050355]. Source data are provided with this paper.

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

## Acknowledgements
We wish to thank S. Miyamoto-Kohno for the insect cell expression. All cryo-EM data in this study were collected at the cryo-EM facility of the RIKEN Center for Biosystems Dynamics Research (Yokohama, Japan). We thank H. Ehara, T. Uchikubo, and R. Akasaka for their help with cryo-EM data collection and analysis. This work was supported by grants from JSPS/MEXT KAKENHI (JP22H02564 to A.Y.). This work was partially supported by the Research Support Project for Life Science and Drug Discovery (Basis for Supporting Innovative Drug Discovery and Life Science Research (BINDS)) from AMED under Grant Number JP23ama121027 to T.T. (support number 4529), RIKEN BDR Structural Cell Biology Project to M.S., RIKEN Pioneering Projects Biology of Intracellular Environment to M.S., RIKEN Pioneering Projects Long-time scale molecular chronobiology to M.S.

## Author contributions
A.Y. conceived the study. A.Y. and K.I. performed sample preparation, biochemical analyses, and cryo-EM data collection. A.Y. performed cryo-EM data processing and structural modeling. T.S. and N.D. performed mass spectrometry analysis. T.T. performed MD simulations. M.S. helped with the sample preparation and the structural analysis. A.Y. wrote the manuscript, which T.S. and T.T. edited. All authors analyzed and discussed the results and contributed to the manuscript.

## Competing interests
The authors declare no competing interests.
