## [Peer Review File · Nature Communications]

Reviewers' Comments:

Reviewer #1:

Remarks to the Author:

This manuscript by Yamagata et al. describes the long-awaited full-length structure of human SV2A. I know of several research groups that have attempted to obtain this important structure and previously failed. The structure presents complexes both with the Botulinum neurotoxin and with antiepileptic drugs, both highly interesting aspects of SV2A. The structures are of very good quality and an impressive amount of high-quality work is presented. No additional experiments are needed. However, several aspects of the manuscript need to be clarified and some analysis need to be added. Furthermore, great experimental results are somewhat clouded by an unwarranted proposal of an alternative "uptake" mechanism. Detailed comments follow below.

It is now always clear what structure you refer to in the results section - particularly when talking about the SV2A overall structure +/- LEV or +/-SV2A - please clarify throughout.

BoNT/A and A1 vs A2 are used carelessly. Please specify everywhere if you refer to A1 (used clinically) or A2. The SV2 binding and the ganglioside is highly similar between the clinically relevant A1 and the A2 variant. This should be clarified, exemplified and referenced accordingly.

The following sentence mixes up, BoNT/A and BoNT/E and SV2C and SV2A. This needs to be clarified and references need to be added updated accordingly.

"Structural analyses using the isolated LD4 domain of SV2C, and the fusion protein of SV2A LD4 and SV2C LD4, revealed that BoNT HC recognizes SV2 LD4 through backbone-backbone and protein-glycan interactions"

Regarding Figure 2f: Would the rearrangements required for the simultaneous binding of BoNT/E to SV2A and GD1a be sterically possible? Perhaps a very short MD or crude manual repositioning could answer this?

Why is LEV binding specific to SV2A. Are the residues in the LEV binding pocket conserved between SV2A, SV2B and SV2C or not?

A sequence alignment between human SV2A, SV2B and SV2C with key residues (both LEV and BoNT) highlighted would be a very useful addition to the supplemental information.

The authors introduce a strange model for how they think SV2A carries out transport (without structural movements between inwards and outwards facing conformations?). How do the authors explain that galactose can be taken up when SV2A by their vesicle recycling mechanism, when it is expressed in yeast, and transports enough galactose to sustain growth? The proposed mechanism cannot explain what happens in yeast. Also, in my opinion, the helices will be able to "bend" to utilize the classical uptake mechanism used by e.g. Glut3. The manuscript would benefit from removing this entire speculation, the results are in general really good and interesting. The speculations regarding the "static conformation" uptake mechanism only hurts the manuscript.

SV2 sees very big changes in pH, please comment on how this could affect binding and conformations of SV2A.

How would galactose fit in the active site? The authors reference a publication that describes that it can be transported.

Material and methods:

Was it full-length that was expressed. Please include aminoacidic range (even if it is full length), since different splice forms exist.

Reviewer #2:

Remarks to the Author:

The manuscript from Yamagata et al. presents an extensive and impressive structural work on the receptor-recognition mechanism of the botulinum neurotoxin, as well as on the target-recognition one of common anti-epileptic drugs used in clinics. To determine the structures of the integral membrane protein SV2A, an essential protein found in the synaptic vesicles, in complex with the anti-epileptic approved drugs, the authors took an elegant and informative approach: they used the botulinum neurotoxin receptor-binding domain as a fiducial binder to confer molecular features to SV2A and enables single-particle cryo-EM analysis. In this way, they “kill two birds with one stone”: one side, they unraveled the interactions of the anti-epileptic drugs commonly used in the clinics with SV2A and proposed a novel uptake mechanism. On the other side they uncover the receptor-recognition mechanism of the botulinum neurotoxin for the first time in the context of the FULL-LENGTH receptor. Previous structural work on that mechanism was done with the isolated SV2 extracellular domain.

The structures presented in the manuscript are of good quality, and will be of interest to a broad readership in several fields, including neuroscience, microbiology, and membrane transport. The figures are carefully made, and represent well the results and claims on the manuscript. However, I have several comments on the text that the authors might want to consider for a revised manuscript:

-The introduction lacks a clear and general description of the structural work published on SV2(A,B, or C), and their complexes with anti-epileptic drugs, or the botulinum neurotoxin. Please, clearly state what has and has not been done. The abstract would also benefit from a brief sentence stating the lack of structural data on the SV2A+antiepileptic-drug complexes, and on full-length receptor+ botulinum neurotoxin ones. If the introduction becomes too long, I would suggest to remove the part on the SYT1 complexes, lines 51-78. Those interactions are very interesting but certainly out of the scope of the current work. Although in methods it is mentioned that SYT1 was co-expressed with SV2A, the SV2A-SYT1 complex couldn't be purified due to poor expression of SYT1.

Also, in the introduction, please describe briefly the fold of the LD4 domain (line 57).

-Results. Line 113. It is unclear if the SV2A-HcA2 2:2 complex has LEV bound. More importantly, the authors state that dimerization occurs through LD4-LD4 interactions, but that the dimer is unlikely to be “physiologically relevant”. How are the authors so certain about the physiological relevance of the dimer? Could it be that the flexibility of the LD4 domain respect to the TM domain, and the curvature of the synaptic vesicle membrane would enable dimer formation? Even a more radical thought, could it be that the SV2A dimers aid determining the curvature of the synaptic vesicles, and that the SV2A dimers play a structural role in vesicle formation? Please, explain in more detail why the dimers are thought to be physiologically irrelevant. It is not immediately obvious from Supp Fig. 4.

-Please, avoid references to the AlphaFold model unless a meaningful comparison between the experimental and predicted models is made. The fact that alpha fold places the cys residues in close proximity does not mean much.

-Regarding glycosylation. The glycosylation pathways in sf9 cells, the system used for the heterologous production of SV2A, differ from those in mammalian or human cells, and efforts have been put to “humanize sf9 cells in that regard (PMID: 12788624). While the authors nicely show the involvement of a complex glycan at the SV2-BoNT interface, they don't discuss the potential differences in glycosylation and their effects on BoNT binding in human cells.

-Line 186, please explain in more detail or add a reference (ref 42?) about the dual-receptor mechanism of BoNT, it is very hard to follow the line of argument in that part of the text.

-Line 220, does the density corresponding to LEV at the bottom of the cavity disappear in the cryoEM maps of the SV2A-BoNT apo complexes?

Line 24, cryoEM structureS and not structure

-Line 126, helices are spread apart, instead of sprayed apart

-Line 369, pellets were resuspended, instead of dissolved

-Line 485, it reads better "selected particles were used for NU"

Reviewer #3:

Remarks to the Author:

The manuscript by Yamagata et al., reports the first structural information for the full length SV2A membrane protein bound to the BoNT receptor-binding domain, BoNT/A2 HC, and the antiepileptic (AED) drugs Levetiracetam (LEV) and Brivaracetam (BRV). The study advances our understanding of the interactions of these molecules with SV2A and appears to be consistent with previous analysis concerning the interaction of these molecules with SV2A and associated disease mutations linked to epilepsy and neurological disorders. The main strengths of the study are the structural data, which reveals important information on the location of the binding site for LEV and BRV, and the role of glycans in the interaction with BoNT/A2. The main weakness is the lack of any binding data to explain some of the differences in LEV and BRV binding. In particular, the authors state that BRV has a 20-fold higher affinity for SV2A compared to LEV. The structures show that BRV does interact differently to LEV in the binding site, in particular with Tyr461 and Ile663. It also appears that these side chains were not analysed in the previous studies. It therefore seems reasonable to support the hypothesis, put forward by the authors that Tyr461 and Ile663 are responsible for the higher binding affinity, with binding data. Have the authors tried nanoDSF or SPA? The addition of this data would strengthen the study considerably.

Similarly, the authors state "For further investigation, the C198S and the C583S mutant SV2As were overexpressed and purified (Extended Data Fig. 6). Both mutant proteins showed expression levels and size exclusion chromatography profiles comparable to the wild-type protein. Therefore, we concluded that SV2A does not form a disulfide bond between Cys198 and Cys583." I don't agree with this conclusion. The nature of biochemistry is that it only informs you when you observe something; it cannot be used to rule out the absence of an observation. I would suggest either repeating the MS using slightly oxidising conditions (CuPh) or, saying that the protein used in this study appears to not contain disulphide bond, although one cannot rule out the formation of such an interaction in the function of SV2A.

Overall the study reports the structure of an interesting SV protein, which will be read with interest by several fields. The lack of any biochemical data to support the structural analysis reduces the impact of the study.

Re: NCOMMS-23-31376A

We are grateful to the reviewers for their helpful comments and made every effort to improve the manuscript according to their comments. We hope that our edits and responses we provide below satisfactorily address all the issues and concerns the reviewers have noted.

Reviewer #1:

1. It is now always clear what structure you refer to in the results section - particularly when talking about the SV2A overall structure +/- LEV or +/-SV2A – please clarify throughout.

We apologize that the description of the structure was not satisfactory. We have revised the introduction to describe which structure was determined in this study, as follows;

“...we determined the cryo-electron microscopy (cryo-EM) structures of human SV2A in complex with LEV (SV2A–LEV), that in complex with BoNT/A2 H_C (SV2A–H_CA2), that in complex with both LEV and BoNT/A2 H_C (SV2A–H_CA2–LEV), and that in complex with both BRV and BoNT/A2 H_C (SV2A–H_CA2–BRV).” (Page 5, line 91)

We have used these terms throughout the text. All the structures have been clarified whether they are in complex with LEV/BRV or not throughout the text (e.g. Page 6., lines 107 and 114, etc.).

2. BoNT/A and A1 vs A2 are used carelessly. Please specify everywhere if you refer to A1 (used clinically) or A2. The SV2 binding and the ganglioside is highly similar between the clinically relevant A1 and the A2 variant. This should be clarified, exemplified and

referenced accordingly.

Thank you for pointing this out. We have briefly described the subtypes of BoNT/A, and BoNT/B in the introduction (Page 4, Line 67). We have clarified A1 or A2 throughout the text (e.g. Page 4, line 73, Page 10, line 188., etc).

3. The following sentence mixes up, BoNT/A and BoNT/E and SV2C and SV2A. This needs to be clarified and references need to be added updated accordingly.

“Structural analyses using the isolated LD4 domain of SV2C, and the fusion protein of SV2A LD4 and SV2C LD4, revealed that BoNT HC recognizes SV2 LD4 through backbone-backbone and protein-glycan interactions.”

We apologize that our description of the currently available structures of the complexes between SV2 protein and BoNT was confusing. We have revised the text on Page 4, line 81 as follows;

“To date, the crystal structures of the isolated LD4 domain of SV2C (LD4C) and the SV2A LD4:SV2C LD4 fusion protein (LD4AC) have been determined, showing unique beta-helix structures. The structures of the complexes between BoNT/A1 H_C (H_{CA1}) and LD4C in the glycosylated and the unglycosylated states revealed that H_{CA1} recognizes LD4C through backbone-backbone and protein-glycan interactions. The recent structural analysis of the complex between BoNT/E H_C (H_{CE}) and LD4AC exhibited an unexpected binding mode distinct from the H_{CA}–SV2C complex. Importantly, however, no structural information has been available on the interaction between the intact SV2A and BoNT/H_C.”

4. Regarding Figure 2f: Would the rearrangements required for the simultaneous binding of BoNT/E to SV2A and GD1a be sterically possible? Perhaps a very short MD or crude manual repositioning could answer this?

Thank you for pointing this out. We have provided a model of the full-length BoNT/E in complex with SV2A and GD1a by manual repositioning of the LD4 domain in SV2A to satisfy the dual-receptor model, shown in the revised Fig. 2g (Page 10, line 217).

5. Why is LEV binding specific to SV2A. Are the residues in the LEV binding pocket conserved between SV2A, SV2B and SV2C or not?

A sequence alignment between human SV2A, SV2B and SV2C with key residues (both LEV and BoNT) highlighted would be a very useful addition to the supplemental information.

According to the suggestion, we have provided the multiple sequence alignment of SV2 proteins in the revised Extended Data Fig. 5. The residues involved in the LEV/BRV binding site are almost completely conserved. We have discussed it on Page 11, line 229 as follows; “The residues involved in the LEV-binding are perfectly conserved in SV2B and SV2C with two exceptions that Ile273 and Cys297 in SV2A are substituted with Leu and Gly in SV2B, respectively (Extended Data Fig. 5). These two substitutions may reduce the binding ability of SV2B to LEV. However, the highly conserved LEV-binding site in SV2 proteins is inconsistent with the strict specificity of LEV to SV2A, suggesting an unknown modulation mechanism in SV2B and SV2C.”

6. The authors introduce a strange model for how they think SV2A carries out transport (without structural movements between inwards and outwards facing conformations?). How do the authors explain that galactose can be taken up when SV2A by their vesicle recycling mechanism, when it is expressed in yeast, and transports enough galactose to sustain growth? The proposed mechanism cannot explain what happens in yeast. Also, in my opinion, the helices will be able to “bend” to utilize the classical uptake mechanism used by e.g. Glut3. The manuscript would benefit from removing this entire speculation, the results are in general really good and interesting. The speculations regarding the “static conformation” uptake mechanism only hurts the manuscript.

According to the suggestion, we have removed the description regarding the “static conformation” uptake mechanism. Instead, we have proposed a new transport mechanism with a conformational change driven by the low internal pH of SV. We have removed the section “Structural comparison with glucose transporters” and have provided a new section “Putative proton-driven transport mechanism” on Page 13, line 278, and Figure 5.

7. SV2 sees very big changes in pH, please comment on how this could affect binding and conformations of SV2A.

Thanks for pointing this out. We have provided a new transport model driven by the low internal pH of SV, as described above (Page 14, line 295).

8. How would galactose fit in the active site? The authors reference a publication that describes that it can be transported.

We manually docked galactose into SV2A and found that the LEV-binding pocket was large enough to accommodate the galactose. We have described it on Page 15, line 319 as well as in the revised Extended Data Fig. 20.

9. Material and methods:

Was it full-length that was expressed. Please include aminoacidic range (even if it is full length), since different splice forms exist.

According to the suggestion, we have described the amino acid range of the full-length SV2A used in this study as follows;

“The full-length SV2A (UniprotID: Q7L0J3 residues 2–742) fused with N-terminal FLAG tag was overexpressed in ...”(Page 6, line 101)

Reviewer #2:

1. The introduction lacks a clear and general description of the structural work published on SV2(A,B, or C), and their complexes with anti-epileptic drugs, or the botulinum neurotoxic. Please, clearly state what has and has not been done.

We apologize that our description of the previously determined structures was confusing.

We have revised the introduction as follows;

“To date, the crystal structures of the isolated LD4 domain of SV2C (LD4C) and the SV2A LD4:SV2C LD4 fusion protein (LD4AC) have been determined, showing unique beta-helix

structures. The structures of the complexes between BoNT/A1 H_C (H_{CA1}) and LD4C in the glycosylated and the unglycosylated states revealed that H_{CA1} recognizes LD4C through backbone-backbone and protein-glycan interactions. The recent structural analysis of the complex between BoNT/E H_C (H_{CE}) and LD4AC exhibited an unexpected binding mode distinct from the H_{CA}-SV2C complex. Importantly, however, no structural information has been available on the interaction between the intact SV2A and BoNT/H_C.” (Page 4, line 81)

The abstract would also benefit from a brief sentence stating the lack of structural data on the SV2A+antiepileptic-drug complexes, and on full-length receptor+ botulinum neurotoxic ones.

According to the suggestion, we have briefly described the lack of structural data on the full-length SV2A in complex with AEDs and BoNT H_C in the abstract, as follows;

“Nevertheless, no structural analysis on the AEDs and BoNT recognition by full-length SV2A has been available.” (Page 2, line 24)

If the introduction becomes too long, I would suggest to remove the part on the SYT1 complexes, lines 51-78. Those interactions are very interesting but certainly out of the scope of the current work. Although in methods it is mentioned that SYT1 was co-expressed with SV2A, the SV2A-SYT1 complex couldn't be purified due to poor expression of SYT1.

According to the suggestion, we have revised the description of SYT1, as follows;

“SYT1 is a Ca²⁺ sensor during SV exocytosis, and therefore the interaction between SV2A and SYT1 should play an important role in modulating the Ca²⁺-dependent SV release. However, this may explain only part of the SV2A function, because the SV2A mutant lacking its N-terminal cytoplasmic region can restore normal synaptic transmission in neurons cultured from SV2A/B DKO mice.” (Page 3, line 60)

Also, in the introduction, please describe briefly the fold of the LD4 domain (line 57).

We have briefly described the fold of the LD4 domain as follows;

“To date, the crystal structures of the isolated LD4 domain of SV2C (LD4C) and the SV2A LD4:SV2C LD4 fusion protein (LD4AC) have been determined, showing unique beta-helix structures.” (Page 4, line 81)

2. Results. Line 113. It is unclear if the SV2A-HcA2 2:2 complex has LEV bound.

We apologize that our description of the dimeric assembly of the SV2A-HcA2-LEV complex was confusing. We have revised the text, as follows;

“In addition to the monomeric SV2A-HcA2-LEV complex, we also obtained the cryo-EM reconstruction of the dimeric assembly of the SV2A-HcA2-LEV complex to 2.90 Å resolution,…”(Page 8, line 113)

More importantly, the authors state that dimerization occurs through LD4-LD4 interactions, but that the dimer is unlikely to be “physiologically relevant”. How are the authors so certain about the physiological relevance of the dimer? Could it be that the flexibility of the LD4

domain respect to the TM domain, and the curvature of the synaptic vesicle membrane would enable dimer formation? Even a more radical thought, could it be that the SV2A dimers aid determining the curvature of the synaptic vesicles, and that the SV2A dimers play a structural role in vesicle formation? Please, explain in more detail why the dimers are thought to be physiologically irrelevant. It is not immediately obvious from Supp Fig. 4.

We are grateful for the very interesting idea that the dimeric SV2A determines the curvature of SV. However, we also must be cautious, because the previous electron microscopy observation of the synapse from the SV2A/B or SV2A knock-out mice did not alter the size, number, and location of SV {Crowder et al. 1999; Janz et al. 1999}. We have discussed it in the revised “Discussion” section (Page 16, line 347).

3. Please, avoid references to the Alphafold model unless a meaningful comparison between the experimental and predicted models is made. The fact that alpha fold places the cys residues in close proximity does not mean much.

We have removed the description of the AlphaFold model (Page 7, line 135).

4. Regarding glycosylation. The glycosylation pathways in sf9 cells, the system used for the heterologous production of SV2A, differ from those in mammalian or human cells, and efforts have been put to “humanize sf9 cells in that regard (PMID: 12788624). While the authors nicely show the involvement of a complex glycan at the SV2-BoNT interface, they don’t discuss the potential differences in glycosylation and their effects on BoNt binding in human cells.

Thanks for pointing this out. We have discussed the potential difference in glycosylation between insect cells and human cells (Page 9, line 173). In brief, the core sugars including two NAGs and a Fucose, which are responsible for targeting by BoNT, are involved in both insect cells and mammalian cells.

5. Line 186, please explain in more detail or add a reference (ref 42?) about the dual-receptor mechanism of BoNT, it is very hard to follow the line of argument in that part of the text.

We have revised the text on Page 10, line 195 including appropriate references, as follows; “Most BoNT H_{Cs} can bind to gangliosides, which are rich on the neuronal cell surface, and simultaneously bind to their corresponding protein receptor, forming a dual receptor complex{Montecucco, 2004}. The structures of the BoNT/A H_{Cs} from the different subtypes in complex with ganglioside revealed that they form a common binding pocket containing a conserved SxWY motif{Gregory, 2023}.”

6. Line 220, does the density corresponding to LEV at the bottom of the cavity disappear in the cryoEM maps of the SV2A-BoNT apo complexes?

We have described the density at the bottom of the cavity in the cryo-EM maps of the apo-SV2A-H_CA2 complex, on Page 11, line 229. In brief, we observed an unidentified density, but it is apparently different from that of LEV. We have also shown it in the revised Extended Data Fig. 16.

7. Line 24, cryoEM structureS and not structure

Thanks for the correction. We have revised it (Page 2, line 26).

8. Line 126, helices are spread apart, instead of sprayed apart

Thanks for the correction. We have revised it (Page 7, line 129).

9. Line 369, pellets were resuspended, instead of dissolved

Thanks for the correction. We have revised it (Page 19, line 394).

10. Line 485, it reads better “selected particles were used for NU”

According to the suggestion, we have revised it (Page 24, line 508).

Reviewer #3:

1. It therefore seems reasonable to support the hypothesis, put forward by the authors that Tyr461 and Ile663 are responsible for the higher binding affinity, with binding data. Have the authors tried nanoDSF or SPA? The addition of this data would strengthen the study considerably.

We thank the reviewer for the great suggestions. After our efforts on nanoDSF, Surface plasmon resonance assay, etc., we finally succeeded in measuring the binding affinity

between SV2A and BRV using the label-free spectral shift assay of the intrinsic tryptophan fluorescence at 350 nm/330 nm. We verified the lower binding affinity of the Y461A or I663A mutant SV2As than that of the wild-type protein, consistent with the structure. We have described these results on Page 13, line 257, and in the revised Fig. 4. We have briefly described it in the abstract (Page 2, line 30), and have also provided the detailed method of this new technique on Page 26, line 568.

2. ...I don't agree with this conclusion. The nature of biochemistry is that it only informs you when you observe something; it cannot be used to rule out the absence of an observation. I would suggest either repeating the MS using slightly oxidising conditions (CuPh) or, saying that the protein used in this study appears to not contain disulphide bond, although one cannot rule out the formation of such an interaction in the function of SV2A.

According to the suggestion, we revised the text on Page 7, line 143, as follows;

“Therefore, the SV2A used in this study appears not to contain a disulfide bond, although one cannot rule out the disulfide bond formation depending on the redox condition of SV in neuronal cells.”

Reviewers' Comments:

Reviewer #1:

Remarks to the Author:

The authors have made a big effort and have addressed all of my questions and concerns. This is an important study that will be highly relevant for both the studies of the Botulinum neurotoxins receptor binding and the understanding and further development of antiepileptic drugs.

Best regards,
Pål Stenmark

Reviewer #2:

Remarks to the Author:

The authors have done a careful work during the revision and I have no further concerns. I recommend publication and congratulate the authors on their work.

Reviewer #3:

Remarks to the Author:

The authors have made some positive changes to their manuscript in light of the previous review comments. The addition of the binding assay shown in Fig. 4 certainly improves the interpretation of the structural data and supports the binding mode proposed for LEV and BRV. However, I do have some additional comments that I think the authors should consider and discuss with the editor.

- The section on proton coupling (lines 278-306) is really a hypothesis rather than a mechanism. The former implies a set of ideas, which are provided around the role of E182 and E262, etc., while the latter implies an explanation of experimental results. As far as I could tell from the current draft, the authors have no evidence for the role of the side chains they highlight in this section. If this is the case, then I would either provide support for this hypothesis through the use of assay data, move this section to the discussion and Figure 5 to the Supplementary Information section and make clear this is a current hypothesis, or remove it altogether from the draft.

- Galactose binding. Why not just test whether galactose binds using your new assay? I think this would make a very nice addition to the data and also enable a more definitive statement from the authors.

Overall, this study reports several new and insightful structures of SV2A bound to clinically relevant drugs and neurotoxins. The study in its present form does not, in my opinion, provide any insight into the mechanism of transport by SV2A. However, I do not see this as a major weakness. Although, the study does need redrafting to take this into consideration.

Re: NCOMMS-23-31376B

We sincerely appreciate the reviewers' positive and constructive comments and their efforts to carefully review our manuscript. Below is our point-by-point response to each comment.

Reviewer #1:

The authors have made a big effort and have addressed all of my questions and concerns. This is an important study that will be highly relevant for both the studies of the Botulinum neurotoxins receptor binding and the understanding and further development of antiepileptic drugs.

We greatly appreciate the reviewer's positive comment.

Reviewer #2:

The authors have done a careful work during the revision and I have no further concerns. I recommend publication and congratulate the authors on their work.

We greatly appreciate the reviewer's positive comment.

Reviewer #3:

1. The section on proton coupling (lines 278-306) is really a hypothesis rather than a mechanism. The former implies a set of ideas, which are provided around the role of E182

and E262, etc., while the latter implies an explanation of experimental results. As far as I could tell from the current draft, the authors have no evidence for the role of the side chains they highlight in this section. If this is the case, then I would either provide support for this hypothesis through the use of assay data, move this section to the discussion and Figure 5 to the Supplementary Information section and make clear this is a current hypothesis, or remove it altogether from the draft.

According to the suggestion, we have moved the section on proton coupling from Results to Discussion (lines 304 – 337). In addition, we have changed the phrase “proposed...mechanism” to “hypothesized...model” (line 321). Figure 5 has moved to the revised Extended Data Fig. 22. The previous studies showed that the D179A and the E182A mutant proteins do not affect the binding to racetam derivatives (Shi et al. 2011 Biochem Soc Trans). The D179A/E182A or the D179N/E182Q mutant proteins were reported to disrupt the synaptic localization of SV2A (Chang & Sudhof 2009; Nowack et al. 2010). These suggest that the interhelical hydrogen bonds (Asp179–Ile269 and Glu182–Arg262) play an important role in protein folding and loss of these hydrogen bond may trigger the conformational transition for transport. We have revised the manuscript including these previous mutational studies (lines 321–335).

2. Galactose binding. Why not just test whether galactose binds using your new assay? I think this would make a very nice addition to the data and also enable a more definitive statement from the authors.

We thank the reviewer for the great suggestions. We tested whether SV2A binds to galactose using our spectral shift assay. We could not detect any significant change in the spectral shift. Therefore, it remains elusive if SV2A directly transports galactose. We have added these contents in the revised manuscript (lines 294–303).